# LLM-GC: Temporal-Semantic Disentanglement with Retrieval Augmentation to Activate LLM's Ability for Multimodal Granger Causal Discovery

## Abstract

Recent advances in neural Granger causal methods have shown promise in modeling temporal nonlinear dependencies. However, existing approaches remain confined to raw time-series data, inherently lacking contextual semantics and tending to overfit, which undermines their real-world applicability. To address these challenges, we propose **LLM-GC**, a novel LLM-empowered multimodal Granger causality discovery framework that enriches unimodal temporal dynamics with semantic priors and world knowledge distilled from large language models (LLMs). LLM-GC leverages dual-modality encoding to capture and align both temporal and contextual dynamics by Cross-Modal Dual Retrieval while avoiding causal entanglement across modalities. To extract multimodal causal features, we introduce a causality-aware self-attention mechanism by simply inverting the conventional self-attention structure, enabling a shared causality augmenter to effectively highlight consistent causal patterns across modalities. LLM-GC is the first to bridge LLMs and Granger causality, and experiments on synthetic and real-world benchmark datasets demonstrate that LLM-GC outperforms existing state-of-the-art methods in Granger causal discovery.

## 1 Introduction

Causal discovery from time series (TS) data is a fundamental yet challenging task with wide-ranging applications in fields such as geography (Stein et al., 2025), genetics (Singh et al., 2022), and biology (Yu et al., 2023). Granger causality (GC) (Granger, 1969) is widely adopted for its interpretability and compatibility with modern deep neural networks (DNNs). Neural GC (Tank et al., 2022) models temporal dynamics while regularizing spurious associations. Subsequent work has advanced this area with architectures like RNNs (Khanna & Tan, 2020), Transformers (Kong et al., 2024), and extensions to irregular TS (Cheng et al., 2023) and root cause analysis (Han et al., 2025).

Despite these advancements, existing Granger causal discovery (GCD) methods are fundamentally constrained by the limited expressiveness of raw TS data. By treating temporal TS in isolation, they overlook critical contextual semantics that extend beyond temporal dependencies. For instance, gene expressions specific to the $Y$ chromosome shouldn't facilitate for female samples, and EEG recorded during daytime may follow distinct causal patterns compared to nighttime. Such domain-specific priors, including data collection contexts and policy-related factors, are human-perceived and carry insights for identifying causal relationships. However, it remains inaccessible to unimodal temporal models, making existing methods prone to overfitting narrow causal patterns, amplifying spurious causality, and struggling in data-scarce or structurally underdetermined real-world scenarios.

To address these limitations, we introduce a complementary textual modality that encodes domain priors beyond raw temporal dynamics. By wrapping structured time-series data and associated metadata (e.g., source domain) into descriptive prompts (Jin et al., 2024; Liu et al., 2025a), LLMs inject semantic augmentation into Granger causal discovery. Pretrained on large-scale corpora, LLMs demonstrate strong capabilities in contextual reasoning and generalization (Guo et al., 2025), which can alleviate overfitting and improve performance in low-resource settings, as shown in the experiments section. While recent studies have explored language-informed embeddings in time-series

Figure 1: (a-c) Illustrate the essence of unimodal Granger causal discovery, where LLMs may play a role. (d) LLM-GC: the paradigm for multimodal Granger causal discovery.

forecasting (Zhou et al., 2023; Liang et al., 2024), the current focus primarily lies in aligning time-series modeling with the prompt understanding capabilities of LLMs (Sun et al., 2024; Hu et al., 2025; Liu et al., 2025c). The potential of LLMs to enhance Granger causal discovery remains unexplored. This paper aims to answer two fundamental questions:

(1) *Can LLMs facilitate Granger causal discovery?*

(2) *How to activate capabilities of LLMs to enhance the performance of Granger causal discovery?*

In this paper, we investigate the potential of LLMs in Granger causal discovery through three representative paradigms shown in Fig. 1, trying to answer the two motivating questions. The baseline **TS Encoding** approach (Fig. 1(a)) encodes raw TS via a dedicated temporal encoder. In contrast, the **Fine-tuning LLM for GCD** paradigm (Fig.1(b)) reformulates TS data into textual prompts and fine-tunes LLMs under supervision of ground-truth causal graphs, enabling direct inference through language. The **Prompt Encoding for GCD** paradigm (Fig. 1(c)) instead leverages LLMs as frozen encoders to produce embeddings from prompts for causal inference. To better harness LLM capabilities for GCD, we propose **LLM-GC**, a novel LLM-empowered multimodal Granger causality discovery framework that integrates temporal dynamics with semantic priors and world knowledge distilled from LLMs (Fig. 1(d)). Specifically, LLM-GC adopts a variable-wise dual-modality encoding scheme to jointly capture temporal and contextual dynamics while mitigating inter-modal causal entanglement. A cross-modal dual retrieval module aligns the resulting heterogeneous representations based on semantic similarity. To ensure causal consistency, we further propose a lightweight yet effective causality-aware self-attention (CASA) mechanism that inverts queries, keys, and values to reorient attention towards effect-driven patterns, upon which we construct a shared causality augmenter that highlights modality-invariant causal structures for final graph inference. Our code is in https://anonymous.4open.science/r/LLM-GC, and the main contributions are:

- We are the first to investigate three representative paradigms for integrating LLMs into Granger causal discovery, revealing their potential to enhance causal inference in TS beyond temporal dynamics.

- We propose LLM-GC, a novel LLM-empowered multimodal Granger causality discovery framework that performs variable-wise encoding of temporal and semantic dynamics from dual modalities, and aligns them via cross-modal dual retrieval.

- We introduce a lightweight yet effective causality-aware self-attention (CASA) mechanism that highlights modality-invariant effect-driven patterns by inverting attention components.

- Extensive experiments on five synthetic and real-world benchmark datasets demonstrate that LLM-GC outperforms state-of-the-art GC methods.

## 2 RELATED WORK

### 2.1 GRANGER CAUSAL DISCOVERY IN TIME SERIES

Granger causality (Granger, 1969) is a widely used framework for assessing temporal causal relationships by testing whether one time series improves the prediction of another. Traditional GC based on linear vector autoregressive (VAR) models struggles to capture nonlinear dependencies.

To address this, recent studies have proposed neural GC approaches that leverage the flexibility of deep neural networks. For example, (Tank et al., 2022) uses sparse component-wise networks to infer GC structures, while forecasting-based models improve the interpretability of learned graphs (Zhou et al., 2024; Khanna & Tan, 2020). Others adopt generative models such as dynamic variational autoencoders (Li et al., 2023), or handle irregular and incomplete data (Cheng et al., 2023; 2024). More recent work explores root cause detection via abnormal exogenous signals (Han et al., 2025). However, all these methods operate in a unimodal setting, limiting their ability to incorporate contextual priors and generalize under data scarcity.

## 2.2 Large Language Models for Time Series

Large Language Models (LLMs) have shown strong generalization and reasoning abilities across domains. GPT4TS (Zhou et al., 2023) pioneered the integration of LLMs into time-series forecasting, sparking a new research direction. Subsequent studies further extended LLM applications to TS modeling: (Cao et al., 2024) decomposed series components to enable distribution adaptation; (Chuang et al., 2024) used statistical prompting to boost performance. However, most methods directly feed raw TS into LLMs, ignoring the misalignment between temporal structures and textual representations. To bridge this gap, recent works reprogram TS into textual prompts (Jin et al., 2024), align embedding spaces (Sun et al., 2024), or enable retrieval-based interaction between TS and prompt representations (Liu et al., 2025a). Still, these approaches largely focus on forecasting and modality alignment, overlooking the causal inference objective. To date, no work has explored leveraging LLMs for Granger causal discovery. Our work addresses this gap by introducing a multimodal framework that integrates semantic priors from LLMs into neural GCD.

## 3 Problem Formulation

**Multivariate Time Series**. Consider a complex dynamical system represented by a multivariate time series $\mathbf{X} = \{\mathbf{x}_1, \ldots, \mathbf{x}_T\} \in \mathbb{R}^{T \times N}$. To facilitate localized modeling of temporal dynamics, $\mathbf{X}$ is segmented into a sequence of overlapping or non-overlapping patches $\mathbf{X}_\tau = \{\mathbf{x}_{\tau-L:\tau-1}\} \in \mathbb{R}^{L \times N}$ of fixed length $L$ and stride $s$, with the TS patch index $\tau = \{L+1, \ldots, \lfloor \frac{T-L+s}{s} \rfloor + L\}$.

**Prompt**. We wrap the TS patch $\mathbf{X}_\tau \in \mathbb{R}^{L \times N}$ into prompts $\mathbf{P}_\tau = \{\mathbf{p}_{\tau,1}, \ldots, \mathbf{p}_{\tau,N}\} \in \mathbb{R}^{S \times N}$ along with variables, where $L$ denotes the time lag. Each prompt $\mathbf{p}_{\tau,i}$ has $S$ elements, shown in Fig. 3.

**Granger Causal Discovery**. For a dynamic system, time-series $j$ Granger causes time-series $i$ when the past values of time-series $x_i$ aid in the prediction of the current and future status of time-series $x_j$. Here, we introduce a corresponding textual modality to complement the information missing in the temporal modality, enabling joint guidance for causal discovery. Given input time-series $\mathbf{X}_{\tau-L:\tau-1}$ and propmt $\mathbf{P}_\tau$, we model each sampled variable $x_{t,i}$:

$$x_{\tau,i} = g_i(\mathbf{x}_{<\tau,1}, \mathbf{p}_{\tau,1}, \ldots, \mathbf{x}_{<\tau,N}, \mathbf{p}_{\tau,N}) + \epsilon_{\tau,i} \tag{1}$$

where $\mathbf{x}_{<\tau,i} = \{\mathbf{x}_{\tau-L:\tau-1,1}, \ldots, \mathbf{x}_{\tau-L:\tau-1,N}\}$, $\epsilon_{t,i}$ is an independent noise item, and $g_i(\cdot)$ is a function mapping the past of all the $N$ time series to series $i$. In our dual-modality setting, Granger causality is extended to:

**Definition 1 Multimodal Granger Causal Discovery**. Time series $j$ is Granger non-causal for time series $i$ if for all $(\mathbf{x}_{\tau-L:\tau-1,1}, ..., \mathbf{x}_{\tau-L:\tau-1,N})$, $(\mathbf{p}_{\tau,1}, \ldots, \mathbf{p}_{\tau,N})$ and all $\mathbf{x}'_{\tau-L:\tau-1,j} \neq \mathbf{x}_{\tau-L:\tau-1,j}$ with the corresponding $\mathbf{p}'_{\tau,j} \neq \mathbf{p}_{\tau,j}$:

$$
\begin{aligned}
&g_i(\mathbf{x}_{<\tau,1}, \mathbf{p}_{\tau,1} \ldots, \mathbf{x}_{<\tau,j}, \mathbf{p}_{\tau,j}, \ldots, \mathbf{x}_{<\tau,N}, \mathbf{p}_{\tau,N}) \\
=&g_i(\mathbf{x}_{<\tau,1}, \mathbf{p}_{\tau,1}, \ldots, \mathbf{x}'_{<\tau,j}, \mathbf{p}'_{\tau,j} \ldots, \mathbf{x}_{<\tau,N}, \mathbf{p}_{\tau,N})
\end{aligned} \tag{2}
$$

i.e., the past data points of time-series $j$ influence the prediction of $x_{\tau,i}$.

## 4 Methodology

To enrich TS causal discovery with semantic and contextual priors, we propose LLM-GC, a multimodal framework that optimizes variable-wise prediction (Eq.1) and infers Granger causality when performance peaks (Eq.2). As shown in Fig. 2, LLM-GC includes three modules: dual-modality encoding, cross-modal retrieval alignment, and causal graph discovery.

Figure 2: Overview of the LLM-GC framework for multimodal Granger causal discovery, which integrates variable-wise dual-modality encoding to capture both temporal and semantic dynamics, cross-modal dual retrieval to align them, and a causality augmenter for refined causal inference.

## 4.1 VARIABLE-WISE DUAL-MODALITY ENCODING

### 4.1.1 TIME SERIES ENCODING BRANCH

The time series branch employs an inverted embedding (Liu et al., 2024a), which defines the patch time series of a variable as a token to effectively capture complex temporal dependencies between these tokens. We invert and embed each patch $\mathbf{X}_\tau = \{\mathbf{x}_{\tau-L:\tau-1}\} \in \mathbb{R}^{L \times N}$ into $\mathbf{E}_\tau = \{\mathbf{e}_{\tau,1}, \ldots, \mathbf{e}_{\tau,N}\} \in \mathbb{R}^{N \times D}$ by performing variable-wise dimensional mapping from $L$ to $D$.

**Time Series Encoder.** Following the *causality augmenter* (see details in the Causal Graph Discovery section), the TS embeddings $\mathbf{E}_\tau$ are passed into a lightweight encoder based on the Multihead Self Attention mechanism (Vaswani et al., 2017), denoted as $MHSA(\cdot)$. For the input $\mathbf{E}_\tau^{(l)}$ at the $l$-th $MHSA(\cdot)$, we have the operation to capture the temporal dependencies of variables defined as:

$$\overline{\mathbf{E}}_\tau^{(l)} = MHSA(\mathbf{E}_\tau^{(l)}) = Concat(\widetilde{\mathbf{E}}_1^{(l)}, \ldots, \widetilde{\mathbf{E}}_K^{(l)})\boldsymbol{w}_O^{(l)}, \; \widetilde{\mathbf{E}}_k^{(l)} = \sigma(\mathbf{Q}_k^{(l)}\mathbf{K}_k^{(l)\top}/\sqrt{d_h})\mathbf{V}_k^{(l)} \quad (3)$$

$$\mathbf{Q}_k^{(l)} = \mathbf{E}_\tau^{(l)}\boldsymbol{w}_{Q_k}^{(l)}, \mathbf{K}_k^{(l)} = \mathbf{E}_\tau^{(l)}\boldsymbol{w}_{K_k}^{(l)}, \mathbf{V}_k^{(l)} = \mathbf{E}_\tau^{(l)}\boldsymbol{w}_{V_k}^{(l)} \quad (4)$$

where $\boldsymbol{w}_{Q_k}^{(l)}, \boldsymbol{w}_{K_k}^{(l)}, \boldsymbol{w}_{V_k}^{(l)} \in \mathbb{R}^{D \times d_h}, \boldsymbol{w}_O^{(l)} \in \mathbb{R}^{D \times D}$ are the projection parameters, $d_h = \lfloor \frac{D}{K} \rfloor$, $\widetilde{\mathbf{E}}_k^{(l)}$ is the $k$-th head embedding, $\sigma$ is the activation function, $\overline{\mathbf{E}}_\tau^{(l)}$ represents the intermediate embedding output from $MHSA(\cdot)$ operation.

In our inverted design, layer normalization (Liu et al., 2024b) is applied across features for each variable to stabilize training and retain variable-specific dynamics. A residual connection precedes normalization, and the output is fed into a feed-forward network $FFN(\cdot)$ with another residual connection, completing one encoder layer:

$$\overline{\mathbf{E}}_\tau^{(l+1)} = LN(FFN(\dot{\mathbf{E}}_\tau^{(l)}) + \dot{\mathbf{E}}_\tau^{(l)}), \; \dot{\mathbf{E}}_\tau^{(l)} = LN(\overline{\mathbf{E}}_\tau^{(l)} + \mathbf{E}^{(l)}) \quad (5)$$

$$LN(\overline{\mathbf{E}}_\tau^{(l)}) = \left\{ \left. \frac{\mathbf{e}_{\tau,n} - \mathrm{Mean}(\mathbf{e}_{\tau,n})}{\sqrt{\mathrm{Var}(\mathbf{e}_{\tau,n})}} \right| n = 1, \ldots, N \right\} \quad (6)$$

where $\dot{\mathbf{E}}_\tau^{(l)}$ denotes the intermediate representation after the feed-forward network $FFN(\cdot)$. For brevity, we use $\ddot{\mathbf{E}}_\tau \in \mathbb{R}^{N \times D}$ to denote the final output of the $l$-layer $TSEncoder(\cdot)$.

### 4.1.2 LLM-EMPOWERED PROMPT ENCODING BRANCH

**LLM-Empowered Prompting.** In practice, we identify four key components for constructing effective prompts: (1) dataset context to provide domain-specific background, (2) historical data to preserve temporal continuity, (3) statistical features (e.g., trends, medians) to enhance pattern recognition, and (4) task instruction to guide the transformation of patch embeddings (Fig. 3).

Pre-trained on large-scale multimodal corpora, LLMs acquire broad world knowledge and demonstrate strong language understanding and reasoning capabilities (Brown et al., 2020; Guo et al.,

2025). In our framework, we adopt GPT-2 (Radford et al., 2019) as a frozen backbone to generate prompt embeddings that augment time-series representations. GPT-2 includes a tokenizer and a language model, both kept frozen during training. The tokenizer maps the input prompt $\mathbf{P}_\tau \in \mathbb{R}^{S \times N}$ into tokens, which are then processed by the GPT-2 model to produce prompt embeddings $\mathcal{P}_\tau$. Given the complexity of LLM internals, we abstract the process with a general formulation:

$$\mathcal{P}_\tau = \text{Pre-trained LLM}(\text{Tokenizer}(\mathbf{P}_\tau)) \tag{7}$$

Motivated by the observation that the last token in a prompt captures the most comprehensive information due to the masked self-attention in LLMs (BehnamGhader et al., 2024), we extract the embedding of the last token from each variable's prompt $\mathcal{P}_\tau$ as the LLM's output, denoted by $\widetilde{\mathcal{P}}_\tau = \{\widetilde{\mathbf{p}}_{\tau,1}, \ldots, \widetilde{\mathbf{p}}_{\tau,N}\} \in \mathbb{R}^{N \times M}$, where $M$ is the output dimension of the LLM. A prompt encoder $PEncoder(\cdot)$, structurally mirroring the time-series encoder, is applied for the semantics output $\overline{\mathcal{P}}_\tau$:

> You are an expert in time series representation learning algorithms.
> ----------------------------------------
> **[BEGIN PROMPT]**
> **[Dataset Context]:** <This dataset represents gene expression levels, where the network topologies were from the transcriptional regulatory networks of E. coli and S. cerevisiae. It includes 100 genes over 100 time points. >
> **[Historical Data]:** From $<\tau-L>$ to $<\tau-1>$, the values were $<x_{\tau-L,i}, \ldots, x_{\tau-1,i}>$ every $f$
> **[Statistical Features]:** The overall trend is $<\Delta_{\tau,i}>$, the median is <median_val>...
> **[Task Instruction]:** <Please comprehensively represents the time series>
> **[END PROMPT]**

Figure 3: Prompt example. Elements enclosed in $< \cdot >$ are dynamically instantiated according to the specific properties of the given time series.

$$\widetilde{\mathcal{P}}_\tau = \{\widetilde{\mathbf{p}}_{\tau,n}\mathbf{w}_H \,|\, n = 1, \ldots, N\}, \ \ \overline{\mathcal{P}}_\tau = PEncoder(\widetilde{\mathcal{P}}_\tau) \tag{8}$$

## 4.2 Cross-Modal Dual Retrieval Alignment

To align the time-series and prompt modalities, we introduce a variable-wise bidirectional retrieval module. It uses inverted time-series embeddings $\overline{\mathbf{E}}_\tau^\top \in \mathbb{R}^{D \times N}$ to retrieve relevant prompt embeddings $\overline{\mathcal{P}}_\tau^\top$ (TRP), and vice versa (PRT). This dual-retrieval bridges semantic priors and temporal observations, allowing both modalities to reinforce each other and improve causal inference.

First, we apply a set of linear transformations $\psi_q, \psi_v, \psi_k$ to the time series embeddings $\overline{\mathbf{E}}_\tau$, yielding compact representations: $\psi_q(\overline{\mathbf{E}}_\tau^\top)$, $\psi_k(\overline{\mathbf{E}}_\tau^\top)$, and $\psi_v(\overline{\mathbf{E}}_\tau^\top)$. Similarly, another set of linear layers $\varphi_q, \varphi_v, \varphi_k$ is used to project the prompt embeddings $\overline{\mathcal{P}}_\tau^\top$ into $\varphi_q(\overline{\mathcal{P}}_\tau^\top)$, $\varphi_k(\overline{\mathcal{P}}_\tau^\top)$, and $\varphi_v(\overline{\mathcal{P}}_\tau^\top)$. Next, we compute two variable-wise similarity matrices $\mathbf{M}_\tau^{TRP}, \mathbf{M}_\tau^{PRT} \in \mathbb{R}^{C \times E}$ via scaled dot-product attention followed by softmax with $\otimes$ matrix multiplication:

$$\mathbf{M}_\tau^{TRP} = F_{\text{softmax}}\left(\psi_q\left(\overline{\mathbf{E}}_\tau^\top\right) \otimes \varphi_k\left(\dot{\mathcal{P}}_\tau^\top\right)\right) \tag{9}$$

$$\mathbf{M}_\tau^{PRT} = F_{\text{softmax}}\left(\psi_q\left(\dot{\mathcal{P}}_\tau^\top\right) \otimes \varphi_k\left(\overline{\mathbf{E}}_\tau^\top\right)\right) \tag{10}$$

We perform variable-wise feature aggregation by retrieving information from both modalities using the similarity. Specifically, time series embeddings attend to prompt embeddings via $\mathbf{M}_\tau^{TRP}$, and vice versa via $\mathbf{M}_\tau^{PRT}$. The final output is obtained by fusing the dual retrieval results from both modalities with linear $\omega^{TRP}, \omega^{PRT}$:

$$\ddot{\mathbf{E}}_\tau = \omega_{TRP}\left(\varphi_v(\overline{\mathbf{E}}_\tau^\top) \otimes \mathbf{M}^{PRT}\right) \oplus \psi_q(\overline{\mathbf{E}}_\tau^\top) \tag{11}$$

$$\ddot{\mathcal{P}}_\tau = \omega_{TRP}\left(\varphi_v(\overline{\mathcal{P}}_\tau^\top) \otimes \mathbf{M}^{PRT}\right) \oplus \varphi_q(\overline{\mathcal{P}}_\tau^\top) \tag{12}$$

Through cross-modal dual retrieval alignment, we transfer the knowledge from the pre-trained LLM into time series embeddings, thus improving the model performance.

**Time Series Forecasting.** We design a time-series forecasting module comprising a multivariate Transformer decoder $TSDecoder(\cdot)$, which shares the same architecture as the time-series encoder, followed by a projection function to generate the final prediction:

$$\hat{x}_{\tau,i} = \mathbf{w}_c \cdot TSDecoder\left(\ddot{\mathbf{E}}_\tau^\top + \ddot{\mathcal{P}}_\tau^\top\right) + b_c \tag{13}$$

### 4.3 Causal Graph Discovery

The Causality Augmenter is designed to infer variable-wise Granger causal relationships based on the predictive dynamics captured by the model. Once the model reaches optimal prediction performance, we estimate the causal graph by evaluating the contribution of each source variable to the prediction of a given target variable.

#### 4.3.1 Causal Augmenter

To tackle the increased complexity of causal source identification introduced by the dual-modality setting, we propose a Causality-Aware Self-Attention (CASA) mechanism. CASA is designed to compute attention across variables rather than across time, preserving variable-wise causal interpretability while avoiding information leakage.

Specifically, we transpose the input feature matrix $\mathbf{h} \in \mathbb{R}^{N \times D}$ into $\mathbf{h}^\top \in \mathbb{R}^{D \times N}$ so that each column corresponds to a distinct variable. Unlike conventional self-attention employing projection matrices in $\mathbb{R}^{D \times D}$, CASA replaces them with variable-level projections $\omega_q, \omega_k, \omega_v \in \mathbb{R}^{N \times N}$:

$$\mathbf{q} = \mathbf{h}^\top \omega_q, \ \mathbf{k} = \mathbf{h}^\top \omega_k, \ \mathbf{v} = \mathbf{h}^\top \omega_v \in \mathbb{R}^{D \times N} \tag{14}$$

$$\mathbf{M} = \mathbf{h}^\top \omega_q (\mathbf{h}^\top \omega_k)^\top \in \mathbb{R}^{D \times D} \tag{15}$$

$$CASA(\mathbf{h}) = \mathrm{Softmax}(\mathbf{M}) \mathbf{h}^\top \omega_v \in \mathbb{R}^{D \times N} \tag{16}$$

CASA aligns with the Granger causality paradigm by computing attention across variables. Its projection matrices explicitly encode variable-to-variable influence, enabling direct causal interpretation. Stacking CASA layers forms a Causal Augmenter that captures higher-order dependencies. Unlike prior methods relying on statistical tests or sparsity, CASA introduces three causality-aware projections—query, key, and value—enhancing interpretability and robustness.

**Objective.** The inferred pairwise GC can be represented by an adjacency matrix $\omega_v = \{\omega_v^{:j}\}_{j=1}^N$, where $\omega_v^{:j} \neq 0$ denotes series $i$ Granger causes $j$ and otherwise. This approach has been thoroughly investigated and shows strong empirical support in recent years (Tank et al., 2022; Cheng et al., 2023; Han et al., 2025).

We apply a regularization term on $\omega_q, \omega_k, \omega_v$ to the training loss to promote sparsity in the causal matrix $\omega_v$, improving interpretability.

$$\mathcal{L} = \sum_{\tau=1}^{\frac{T-L+s}{s}} (\hat{x}_{\tau+L,i} - f_{\theta_i}(\mathbf{X}_\tau, \mathbf{P}_t)^2) + \lambda \sum_{j=1}^{N} \left( \parallel \omega_q^{:j} \parallel_2 + \parallel \omega_k^{:j} \parallel_2 + \parallel \omega_v^{:j} \parallel_2 \right) \tag{17}$$

where $\lambda$ is a trade-off between prediction and regularization.

**Optimizing the Penalized Objective.** We use proximal gradient descent (Parikh et al., 2014) to optimize the nonconvex objectives of Eq. 17. Details are in the Appendix.

## 5 Experiments

We evaluate LLM-GC on both synthetic and real-world benchmarks, and conduct ablation studies to assess the impact of modules and different LLM integration strategies. We visualize the performance of LLM-GC across all five benchmark datasets, as illustrated in Fig. 4, where LM-GC consistently achieves superior performance across all five benchmark datasets.

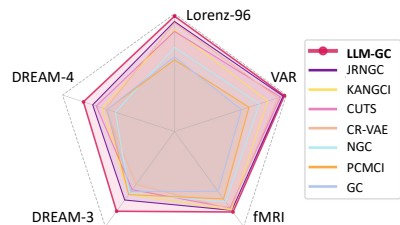

Figure 4: Overall performance of LLM-GC on five benchmarks.

### 5.1 Experimental Setup

**Datasets.** We evaluate the proposed LLM-GC framework on five benchmark datasets. The synthetic datasets are generated from (1) a linear Vector Autoregressive (VAR) model (Tank et al., 2022) and (2) a nonlinear Lorenz-96 model (Karimi & Paul,

2010). The real-world benchmarks include: (3) NetSim (Smith et al., 2011), an fMRI dataset modeling the connectivity dynamics among 15 brain regions; (4) DREAM-3 (Prill et al., 2010) and (5) DREAM-4 (Marbach et al., 2010), two widely used benchmarks for gene regulatory network inference. Dataset Details are provided in the Appendix.

Table 1: Overall performance (mean±std.) on synthetic VAR and Lorenz-96 datasets for Granger causal discovery.

| Synthetic Dataset | Metrics | GC | PCMCI | NGC | CR-VAE | CUTS | KANGCI | JRNGC | LLM-GC |
|---|---|---|---|---|---|---|---|---|---|
| VAR(20,1000,5) | AUROC (↑) | 0.598±0.020 | 0.666±0.020 | 0.759±0.020 | 0.925±0.015 | 0.947±0.010 | 0.833±0.025 | 0.970±0.019 | **0.982±0.033** |
| | AUPRC (↑) | 0.605±0.014 | 0.743±0.017 | 0.745±0.015 | 0.965±0.018 | 0.980±0.032 | 0.840±0.015 | 0.970±0.020 | **0.989±0.018** |
| | F1 (↑) | 0.599±0.031 | 0.701±0.028 | 0.733±0.015 | 0.930±0.024 | 0.963±0.015 | 0.852±0.017 | 0.969±0.017 | **0.980±0.017** |
| | SHD (↓) | 38±4 | 33±2 | 14±3 | 8±2 | 6±2 | 15±3 | 11±1 | **5±1** |
| VAR(20,500,20) | AUROC (↑) | 0.569±0.020 | 0.598±0.030 | 0.698±0.010 | 0.830±0.013 | 0.842±0.025 | 0.820±0.020 | 0.935±0.015 | **0.971±0.012** |
| | AUPRC (↑) | 0.588±0.025 | 0.587±0.030 | 0.675±0.025 | 0.835±0.015 | 0.837±0.027 | 0.810±0.021 | 0.925±0.016 | **0.973±0.006** |
| | F1 (↑) | 0.708±0.018 | 0.609±0.045 | 0.678±0.002 | 0.810±0.025 | 0.823±0.015 | 0.823±0.005 | 0.946±0.012 | **0.955±0.002** |
| | SHD (↓) | 179±4 | 165±12 | 95±6 | 99±5 | 22±3 | 65±15 | 59±7 | **18±2** |
| VAR(40,1000,20) | AUROC (↑) | 0.599±0.029 | 0.566±0.025 | 0.649±0.015 | 0.785±0.026 | 0.838±0.020 | 0.789±0.015 | 0.919±0.003 | **0.945±0.005** |
| | AUPRC (↑) | 0.578±0.020 | 0.589±0.024 | 0.643±0.017 | 0.845±0.025 | 0.837±0.017 | 0.813±0.021 | 0.902±0.018 | **0.956±0.010** |
| | F1 (↑) | 0.710±0.010 | 0.578±0.032 | 0.638±0.022 | 0.805±0.015 | 0.810±0.029 | 0.790±0.010 | 0.938±0.009 | **0.950±0.002** |
| | SHD (↓) | 164±3 | 158±10 | 85±2 | 83±10 | 79±6 | 58±9 | 33±6 | **10±2** |
| Lorenz(20,1000,10) | AUROC (↑) | 0.633±0.026 | 0.608±0.022 | 0.713±0.020 | 0.923±0.013 | 0.850±0.020 | 0.862±0.018 | 0.934±0.018 | **0.979±0.033** |
| | AUPRC (↑) | 0.610±0.014 | 0.634±0.015 | 0.715±0.028 | 0.893±0.020 | 0.867±0.021 | 0.875±0.009 | 0.946±0.015 | **0.984±0.018** |
| | F1 (↑) | 0.606±0.024 | 0.635±0.010 | 0.728±0.022 | 0.903±0.006 | 0.822±0.019 | 0.873±0.021 | 0.931±0.011 | **0.980±0.017** |
| | SHD (↓) | 48±2 | 42±6 | 29±3 | 9±1 | 14±3 | 12±2 | 10±2 | **8±1** |
| Lorenz(20,500,20) | AUROC (↑) | 0.540±0.018 | 0.575±0.015 | 0.656±0.023 | 0.853±0.020 | 0.813±0.038 | 0.775±0.016 | 0.903±0.020 | **0.943±0.008** |
| | AUPRC (↑) | 0.568±0.010 | 0.586±0.010 | 0.665±0.012 | 0.867±0.018 | 0.862±0.017 | 0.780±0.014 | 0.925±0.022 | **0.950±0.015** |
| | F1 (↑) | 0.690±0.014 | 0.571±0.012 | 0.725±0.013 | 0.565±0.017 | 0.810±0.026 | 0.770±0.010 | 0.915±0.004 | **0.944±0.006** |
| | SHD (↓) | 197±3 | 182±10 | 141±5 | 103±8 | 70±19 | 82±7 | 78±8 | **23±3** |
| Lorenz(40,1000,20) | AUROC (↑) | 0.560±0.019 | 0.557±0.013 | 0.716±0.018 | 0.743±0.021 | 0.825±0.006 | 0.719±0.015 | 0.907±0.008 | **0.932±0.005** |
| | AUPRC (↑) | 0.556±0.010 | 0.543±0.029 | 0.687±0.017 | 0.809±0.017 | 0.829±0.005 | 0.766±0.011 | 0.909±0.017 | **0.940±0.016** |
| | F1 (↑) | 0.571±0.015 | 0.568±0.042 | 0.755±0.010 | 0.768±0.024 | 0.774±0.017 | 0.767±0.029 | 0.913±0.002 | **0.938±0.009** |
| | SHD (↓) | 169±8 | 170±12 | 90±8 | 91±9 | 77±10 | 152±10 | 32±10 | **28±5** |

**Baselines.** We perform comparative experiments with seven competitive methods: GC (Granger, 1969), PCMCI (Runge et al., 2019), NGC (Tank et al., 2022), CR-VAE (Li et al., 2023), CUTS (Cheng et al., 2023), JRNGC (Zhou et al., 2024), KANGCI (Liu et al., 2025b).

**Evaluation Metrics.** We adopt four standard evaluation metrics: (1) AUROC, measuring the area under the ROC curve; (2) AUPRC, capturing the area under the precision-recall curve; (3) F1 Score, the harmonic mean of precision and recall and (4) SHD, Structural Hamming Distance, quantifying differences between predicted and ground-truth.

**Implementation Details.** All experiments were conducted on a server equipped with ten NVIDIA GeForce RTX 3090 GPUs (24 GB memory each). The optimization was performed using the Adam optimizer (Kingma & Ba, 2014) with a CosineAnnealingLR scheduler (Loshchilov & Hutter, 2017), starting from a learning rate of 0.0005 for 1000 epochs. The hyperparameters were tuned through grid search, and the optimal values for all experimental settings are provided in the Appendix.

### 5.2 EXPERIMENT RESULTS ON SYNTHETIC BENCHMARKS

**VAR.** We simulated $N \in \{20, 40\}$ time series over $T \in \{500, 1000\}$ observations with a maximum time lag of $\tau \in \{5, 20\}$. As shown in Table 1, LLM-GC consistently achieves the highest AUROC, AUPRC, and F1 scores, along with the lowest SHD across all VAR settings. Even in the most challenging case ($N = 40, \tau = 20$), LLM-GC surpasses all baselines by a notable margin. Moreover, it shows more stable SHD than JRNGC, showing its robustness and reliability in extracting accurate structures under high-dimensional, long-range dependencies.

**Lorenz-96.** The Lorenz-96 system is used to evaluate model robustness under nonlinear chaotic dynamics. We vary $N \in \{20, 40\}$ and set the forcing constant $F \in \{10, 20\}$, where higher $F$ introduces stronger chaos. LLM-GC consistently outperforms all baselines across all the metrics. Particularly under chaotic regimes (e.g., $F = 20$), traditional and neural methods degrade notably, while LLM-GC maintains strong structure recovery. This superior performance can be attributed to our CASA mechanism, which enables variable-wise causal attention and facilitates interpretable structure learning, also shown in Fig. 6(e).

### 5.3 Experiment Results on Real-world Benchmarks

**fMRI**. We evaluate LLM-GC on the simulated fMRI BOLD dataset, which contains 28 simulations. Each simulation includes time series data from 50 subjects, covering diverse brain connectivity patterns. Unlike previous studies that focused on a limited subset of simulations, we conduct a comprehensive evaluation across all settings. As shown in Fig. 5, LLM-GC achieves competitive AUROC scores in most simulations, and performs favorably in 22 out of 28 cases. While other methods such as KANGCI and JRNGC also show strong results in specific conditions, LLM-GC offers consistent performance with lower variance, suggesting better adaptability across a range of causal patterns. This may be attributed to the incorporation of world knowledge through LLM-based representations, which can help distinguish subtle causal relationships in complex scenarios.

**DREAM-3 and DREAM-4**. We evaluate the performance of LLM-GC on two widely used benchmark datasets for causal discovery from gene expression data: DREAM-3 and DREAM-4 in silico challenges. Each dataset contains five sub-datasets with ground-truth Granger causal graphs. The evaluation metric is AUROC. As shown in Table 2, LLM-GC achieves the highest AUROC scores in all five sub-datasets. Compared to baelines, LLM-GC demonstrates consistently better performance across both bacterial and yeast systems. Table 2 also reports results on the DREAM-4 dataset, where LLM-GC shows leading performance in all five sub-datasets. In contrast, the second-best method JRNGC obtains lower scores in all cases, and classical methods such as GC and NGC show performance degradation, particularly under the limited observation setting of DREAM-4. These results suggest that LLM-GC is competitive across both datasets, including scenarios with complex structures and limited time points.

Table 2: AUROC for the sub-datasets in DREAM-3 and in DREAM-4.

| Models | DREAM-3 | | | | | DREAM-4 | | | | |
|---|---|---|---|---|---|---|---|---|---|---|
| | Ecoli-1 | Ecoli-2 | Yeast-1 | Yeast-2 | Yeast-3 | Gene-1 | Gene-2 | Gene-3 | Gene-4 | Gene-5 |
| GC | $0.557_{\pm0.014}$ | $0.649_{\pm0.010}$ | $0.646_{\pm0.006}$ | $0.623_{\pm0.022}$ | $0.548_{\pm0.003}$ | $0.602_{\pm0.012}$ | $0.502_{\pm0.024}$ | $0.500_{\pm0.007}$ | $0.503_{\pm0.019}$ | $0.514_{\pm0.038}$ |
| PCMCI | $0.6114_{\pm0.023}$ | $0.622_{\pm0.027}$ | $0.637_{\pm0.031}$ | $0.627_{\pm0.001}$ | $0.626_{\pm0.009}$ | $0.603_{\pm0.029}$ | $0.501_{\pm0.035}$ | $0.503_{\pm0.015}$ | $0.510_{\pm0.015}$ | $0.512_{\pm0.005}$ |
| NGC | $0.631_{\pm0.029}$ | $0.629_{\pm0.035}$ | $0.601_{\pm0.015}$ | $0.584_{\pm0.027}$ | $0.592_{\pm0.005}$ | $0.528_{\pm0.031}$ | $0.499_{\pm0.008}$ | $0.489_{\pm0.033}$ | $0.547_{\pm0.011}$ | $0.561_{\pm0.013}$ |
| CR-VAE | $0.652_{\pm0.037}$ | $0.634_{\pm0.004}$ | $0.623_{\pm0.016}$ | $0.590_{\pm0.036}$ | $0.594_{\pm0.021}$ | $0.617_{\pm0.020}$ | $0.524_{\pm0.018}$ | $0.548_{\pm0.025}$ | $0.523_{\pm0.002}$ | $0.569_{\pm0.039}$ |
| CUTS | $0.648_{\pm0.012}$ | $0.568_{\pm0.024}$ | $0.585_{\pm0.007}$ | $0.511_{\pm0.019}$ | $0.531_{\pm0.038}$ | $0.699_{\pm0.014}$ | $0.655_{\pm0.010}$ | $\underline{0.657_{\pm0.006}}$ | $0.643_{\pm0.022}$ | $0.648_{\pm0.003}$ |
| KANGCI | $0.662_{\pm0.017}$ | $0.636_{\pm0.014}$ | $0.641_{\pm0.028}$ | $0.658_{\pm0.037}$ | $0.631_{\pm0.026}$ | $0.649_{\pm0.023}$ | $0.614_{\pm0.027}$ | $0.625_{\pm0.031}$ | $0.637_{\pm0.001}$ | $0.631_{\pm0.009}$ |
| JRNGC | $\underline{0.720_{\pm0.006}}$ | $\underline{0.678_{\pm0.013}}$ | $\underline{0.702_{\pm0.005}}$ | $\underline{0.697_{\pm0.011}}$ | $\underline{0.690_{\pm0.035}}$ | $\underline{0.731_{\pm0.010}}$ | $\underline{0.747_{\pm0.014}}$ | $0.591_{\pm0.000}$ | $\underline{0.642_{\pm0.021}}$ | $\underline{0.655_{\pm0.020}}$ |
| **LLM-GC** | $\mathbf{0.838_{\pm0.024}}$ | $\mathbf{0.780_{\pm0.030}}$ | $\mathbf{0.767_{\pm0.009}}$ | $\mathbf{0.743_{\pm0.037}}$ | $\mathbf{0.762_{\pm0.026}}$ | $\mathbf{0.814_{\pm0.014}}$ | $\mathbf{0.792_{\pm0.012}}$ | $\mathbf{0.713_{\pm0.015}}$ | $\mathbf{0.735_{\pm0.024}}$ | $\mathbf{0.749_{\pm0.018}}$ |

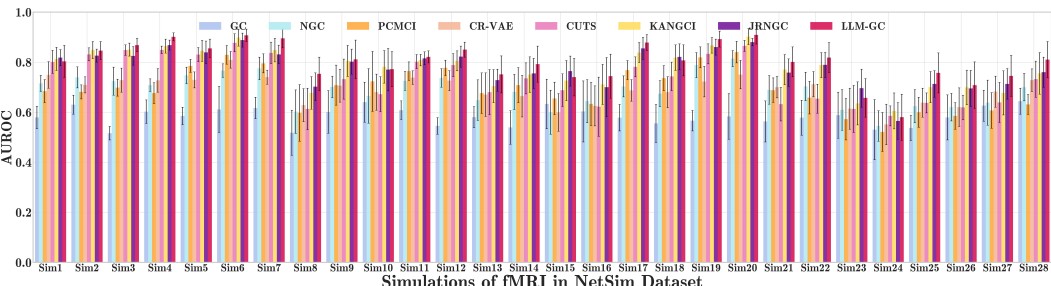

Figure 5: Performance on NetSim Dataset Under AUROC.

### 5.4 Ablaiton Study

**Are LLMs Useful for Granger Causal Discovery?** We implement three representative paradigms: TSE (temporal-only models), Fine-tuning (LLMs trained with causal supervision), Prompt (frozen LLMs used to embed textual prompts), and our Multimodal approach (LLM-GC), which combines time-series signals with LLM-based knowledge retrieval. Figure 6(a-b) reports the ARUOC scores under two datasets. Temporal-only methods serve as a lower bound, with limited capacity to leverage external knowledge. Fine-tuned and prompt-based LLMs show moderate improvements, though performance varies depending on the model and setup. In contrast, LLM-GC achieves the highest ARUOC in both settings, suggesting that combining time series signals with knowledge-enhanced retrieval via LLMs better captures underlying causal relations. Notably, LLM-GC outperforms uni-modal LLM variants, implying that the multimodal integration and structured prompt design in our approach are beneficial for causal discovery tasks.

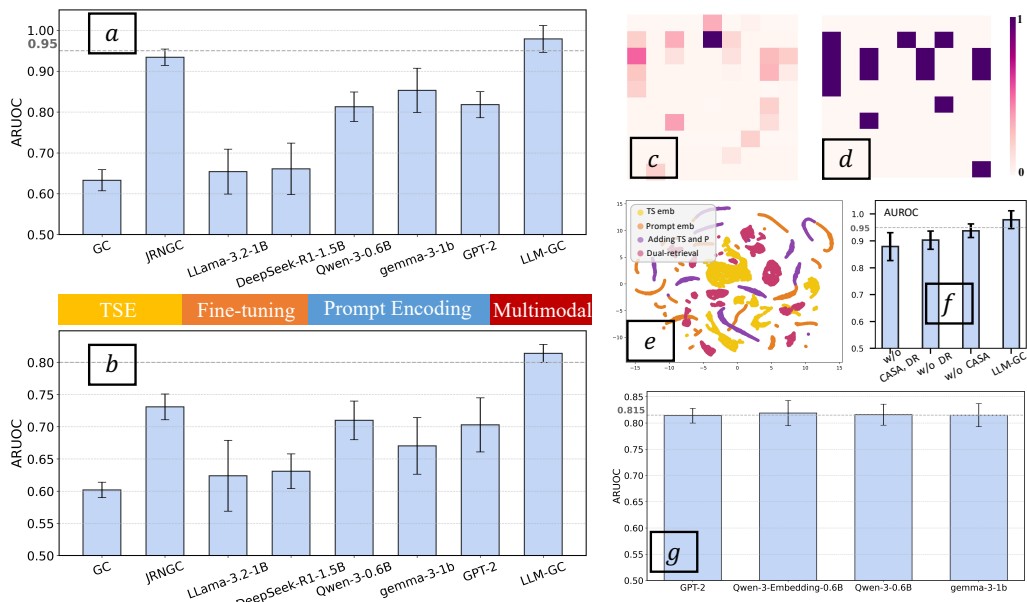

Figure 6: Performance comparison of different paradigms for integrating into Granger causal discovery, on Lorenz-96 (a) and DREAM-4 (b). (c) shows the LLM-GC inferred causality of Ecoli-1 in DREAM-4 and (d) is the ground truth. (e) shows the UMAP visualization of four embeddings. (f) shows ablation of CASA and DR. (g) shows ablation of the frozen LLM.

**Module Ablation.** We conduct an ablation study on the Lorenz-96 dataset to evaluate the contributions of the causality-aware self-attention (CASA) mechanism and the cross-modal dual retrieval (DR) module in LLM-GC. In this study, "w/o" denotes the removal of a specific component. As shown in Fig. 6(f), the complete LLM-GC model achieves the highest AUROC, clearly outperforming its ablated variants. Removing both CASA and DR results in a substantial performance drop, highlighting their combined importance. Excluding DR alone also causes a noticeable decrease in AUROC, indicating that semantic retrieval plays a key role in aligning temporal and contextual representations. Meanwhile, omitting only CASA leads to a moderate decline, suggesting that while cross-modal alignment contributes significantly to capturing informative priors, CASA further refines the causal structure by enforcing variable-wise attention. These findings validate the necessity of both modules in achieving robust causal discovery. We further investigate the impact of different pre-trained LLMs for generating prompt embeddings in the LLM-GC framework, shown in Fig.6(g). While our method is model-agnostic and can accommodate a wide range of LLMs, different models may vary in representation quality, embedding dimensionality, and alignment capability.

**Visualization of Embeddings.** Fig. 6(e) illustrates the distributional patterns of the learned embeddings under different stages. The prompt embeddings exhibit richer inter-variable relationships compared to the TS embeddings. Our dual-retrieval representation forms clearly separated clusters, highlighting improved modality alignment and semantic structure.

## 6    CONCLUSION

In this paper, we explore the potential of Large Language Models to enhance Granger causal discovery from time series data. We compare three representative paradigms of LLM integration: temporal-only models, fine-tuned LLMs, and prompt-based LLMs, and propose LLM-GC, a multimodal framework that incorporates semantic priors and contextual knowledge from LLMs into the causal discovery process. LLM-GC introduces a dual-modality encoder, a cross-modal retrieval module, and a causality-aware self-attention mechanism to align and enhance representations across modalities. Experiments on synthetic and real-world datasets show that LLM-GC consistently outperforms existing GCD methods. Our findings highlight the potential of LLMs as semantic enhancers for causal discovery and suggest new directions for combining language and time series models in scientific and real-world applications.

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

## A APPENDIX

### A.1 MOTIVATIONS AND LIMITATIONS OF THIS STUDY

The motivation of this work is to leverage the semantic priors and world knowledge encoded in large language models (LLMs) to address the limitations of traditional Granger causal discovery, which relies solely on temporal structures. Such structure-only approaches often suffer from overfitting and lack generalizability in real-world scenarios. By integrating contextual semantics from LLMs, we aim to enhance the model's capacity to capture underlying causal mechanisms and improve robustness.

To this end, we propose LLM-GC, a novel multi-modal framework that explicitly aligns the temporal dynamics of time series with the semantic representations derived from LLMs. This design bridges the gap between sequential data and language-based knowledge, enabling more accurate and interpretable Granger causality discovery.

Notably, our focus lies in introducing a new paradigm for causal discovery rather than dissecting the internal mechanisms of LLMs. Experiments are conducted using lightweight LLMs (e.g., 0.1B and

1B parameters) to validate the feasibility of our approach. We emphasize that this study represents an initial step toward LLM-enhanced causal reasoning. Future work will explore larger-scale models and further investigate the potential of LLMs in complex dynamical systems.

## A.2 BACKGROUD

### A.2.1 GRANGER CAUSALITY

Granger causality (Granger, 1969; Dahlhaus & Eichler, 2003) is a widely used framework for modeling causal relationships in multivariate time series. The core idea is that if the prediction of a target variable $x_j$ can be significantly improved by incorporating the past values of another variable $x_i$, then $x_i$ is said to "Granger cause" $x_j$. While the original formulation of Granger causality assumes linear relationships, recent advances have extended it to capture nonlinear dependencies (Tank et al., 2022; Assaad et al., 2022).

Given a multivariate time series $\mathbf{X} = \{\mathbf{x}_1, \ldots, \mathbf{x}_T\} \in \mathbb{R}^{T \times N}$ with $N$ variables and $T$ time steps, the temporal evolution of variable $x_j$ is modeled as:

$$x_j(t) = f_j(x_1(< t), \ldots, x_N(< t)) + \epsilon_j, \tag{18}$$

where $x_k(< t)$ denotes the historical observations of variable $x_k$ before time $t$, and $\epsilon_j$ is an independent noise term. The function $f_j$ maps the historical context of all variables to the future value of $x_j$.

If adding the past values of variable $x_i$ to the input of $f_j$ leads to a statistically significant improvement in the prediction of $x_j(t)$, then $x_i$ is considered a Granger cause of $x_j$. In this sense, Granger causality identifies the parent set of each variable as those that provide predictive information for its future values.

**Limitations of Granger Causality.** While Granger causality provides a valuable framework for identifying temporal causal dependencies, it is crucial to recognize its underlying assumptions and limitations. In particular, it assumes the absence of hidden confounders—i.e., all relevant variables influencing the system are observed and incorporated—and excludes instantaneous effects, requiring that causal influence occurs with a time lag. Violations of these assumptions may result in misleading inferences, underscoring the need for careful validation and the potential consideration of alternative or complementary causal discovery frameworks.

### A.2.2 CAUSAL GRAPH FOR TIME SERIES

Different types of causal graphs can be considered for time series (Assaad et al., 2022). The window causal graph (see Fig. 7(a)) only covers a fixed number of time instants (with a maximum causal influence lag $\tau$ ) and assumes the causal relations amongst different variables are consistent over time. The summary causal graph (see Fig. 7(b)) directly relates variables without any indication of time. Usually, it is difficult to estimate window causal graph because it requires to determine which exact time instant is the cause of another. It is of course easier to estimate a summary causal graph. In practice, it is often sufficient to know the causal relations between time series as a whole, without knowing precisely the relations between time instants (Assaad et al., 2022).

In our work, we consider recovery of a Granger causal graph (see Fig. 7(c)), which separates past observations and present values of each variable and aims to if the past of $\mathbf{x}_i$ (denoted $\mathbf{x}_{<t,i}$) causes the present value of $\mathbf{x}_j$ (denoted $\mathbf{x}_{<t,j}$ ). Obviously, our Granger causal graph lies between the window causal graph and the summary causal graph.

## A.3 SUPPLEMENTARY DETAILS OF THE LLM-GC FRAMEWORK

### A.3.1 CAUSAL GRAPH CONSTRUCTION

Let $G = (V, E)$ denote the Granger causal graph, where $V$ is the set of nodes representing $N$ dependent time series $\mathbf{x}_1, \ldots, \mathbf{x}_N$, and $E$ is the set of directed edges that capture the underlying causal relationships.

To model the dynamics of each target time series $\mathbf{x}_i$, we construct a dedicated prediction function $f_{\theta_i}$, which learns to approximate the optimal predictive mechanism for $\mathbf{x}_i$ based on the historical

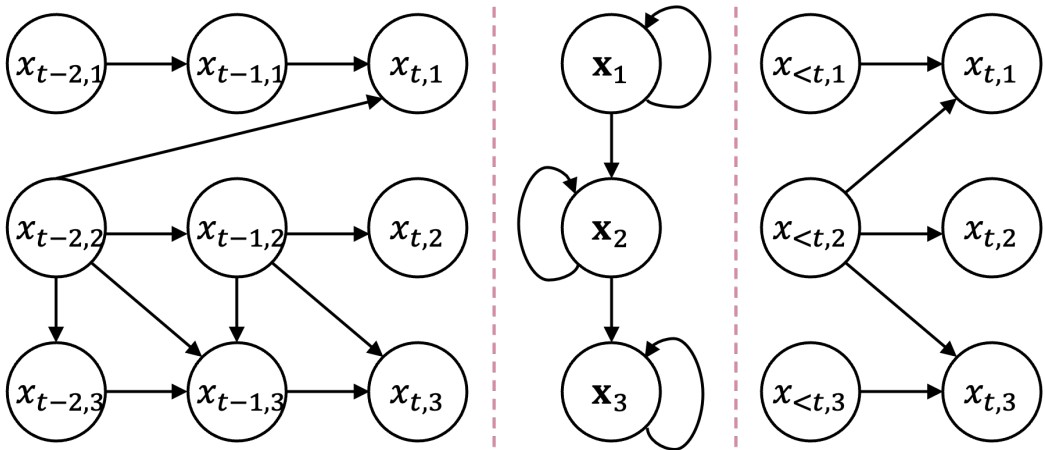

Figure 7: Example of (a) window causal graph; (b) summary causal graph; and (c) our Granger causal graph. Figure adapted from (Li et al., 2023).

information of all other time series. During training, each $f_{\theta_i}$ produces a corresponding causal weight matrix $\omega_{v,i} \in \mathbb{R}^{N \times N}$, encoding the Granger causality structure directed toward $\mathbf{x}_i$.

Collectively, the system yields a set of $N$ causality matrices $\{\omega_{v,1}, \ldots, \omega_{v,N}\}$—one per target variable—each of which captures variable-specific causal dependencies. Specifically, the $j$-th column of $\omega_{v,i}$, denoted as $\omega_{v,i}^{:j}$, quantifies the causal contribution from time series $\mathbf{x}_j$ to the prediction of $\mathbf{x}_i$. A higher value in $\omega v, i^{:j}$ indicates a stronger inferred Granger-causal influence from $\mathbf{x}_j$ to $\mathbf{x}_i$.

An edge from node $\mathbf{x}_i$ to $\mathbf{x}_j$ (i.e., $\mathbf{x}_i \rightarrow \mathbf{x}j$) exists if and only if $\alpha_{ij} \neq 0$. Specifically:

- If $i \neq j$, the past values of $\mathbf{x}_i$ provide unique and significant predictive information about $\mathbf{x}_j$.

- If $i = j$, the series exhibits self-causality, meaning $\mathbf{x}_i$ helps predict its own future values.

### A.3.2 Optimizing the Penalized Objective

We use proximal gradient descent (Parikh et al., 2014) to optimize the nonconvex objectives of Eq. (21). This approach is crucial in inducing zeros in input matrix columns, key for interpreting Granger non-causality. A line search might be added to the algorithm to ensure local minimum convergence (Gong et al., 2013).

The algorithm updates the network weights $\boldsymbol{\Theta}$ iteratively starting with $\omega_q, \omega_k, \omega_v$ by

$$\omega_q(i+1) = \text{prox}_\gamma \left( \omega_q(i) - \gamma \nabla \mathcal{L}_{pred}(\omega_q(i)) \right) \tag{19}$$

$$\omega_k(i+1) = \text{prox}_\gamma \left( \omega_k(i) - \gamma \nabla \mathcal{L}_{pred}(\omega_k(i)) \right) \tag{20}$$

$$\omega_v(i+1) = \text{prox}_\gamma \left( \omega_v(i) - \gamma \nabla \mathcal{L}_{pred}(\omega_v(i)) \right) \tag{21}$$

where $\text{prox}_\gamma$ denotes the proximal operator with step size $\gamma$; $\mathcal{L}_{prediction}$ denotes the convex part of the neural network prediction loss.

As the sparsity-promoting group penalties target only the input weights, the proximal step for weights at higher levels simplifies to an identity function. The input weights' group lasso penalty proximal step involves a group soft-thresholding operation (Parikh et al., 2014):

$$\text{prox}_{\gamma\rho}(\omega^{:j}) = \text{soft}(\omega^{:j}, \gamma\rho) = \left( 1 - \frac{\rho\gamma}{||\omega^{:j}||_2} \right)_+ \omega^{:j} \tag{22}$$

where $(x)_+ = \max(0, x)$. Training uses two optimization methods: proximal gradient on input layer weights $\omega_q, \omega_k, \omega_v$, and SGD on other parameters.

---

**Algorithm 1** Proximal gradient descent optimization algorithm.

---

**Require:** $\rho > 0$

$m = 0$, initialize $\theta_i, \omega_q(0), \omega_k(0), \omega_v(0)$ , time series $\mathbf{X}$.

Patching time serires into $\{\mathbf{X}_\tau\}_1^{\lfloor \frac{T-L+s}{s} \rfloor\}}$

Wrapping $\{\mathbf{X}_\tau\}_1^{\lfloor \frac{T-L+s}{s} \rfloor\}}$ into prompts.

Pre-trained LLM processes the prompt and generates the embedding.

**while** not converged **do**

compute $\sum_{\tau=1}^{\frac{T-L+s}{s}} \left( \hat{x}_{\tau+L,i} - f_{\theta_i}(\mathbf{X}_\tau, \mathbf{P}_t)^2 \right) + \lambda \left( \sum_{j=1}^{N} (\| \omega_q^{:j} \|_2 + \| \omega_k^{:j} \|_2 + \| \omega_v^{:j} \|_2) \right)$

compute $\nabla \mathcal{L}_{pred}$ by BPTT and pdate $\Theta$ except $W^0$ using SGD.

$i = i + 1$

determine $\gamma$ by line search.

**for** $j = 1$ to $m$ **do**

$\omega_q^{:j}(i+1) = \text{soft} \left( \omega_q^{:j}(i) - \gamma \nabla_{\omega_q^{:j}} \mathcal{L}_{pred}\left(\omega_q(i)\right), \gamma\rho \right)$

$\omega_k^{:j}(i+1) = \text{soft} \left( \omega_k^{:j}(i) - \gamma \nabla_{\omega_k^{:j}} \mathcal{L}_{pred}\left(\omega_k(i)\right), \gamma\rho \right)$

$\omega_v^{:j}(i+1) = \text{soft} \left( \omega_v^{:j}(i) - \gamma \nabla_{\omega_v^{:j}} \mathcal{L}_{pred}\left(\omega_v(i)\right), \gamma\rho \right)$

**end for**

**end while**

Infer Granger causal graph from $\omega_v$

**return** Granger causal graph

---

### A.3.3 HARMONIZER

**Motivation.** The Harmonizer module is designed to bridge the substantial gap in embedding dimensionality between the time series encoder and the large language model (LLM). In practice, deep neural networks often operate with relatively compact embeddings, typically around 256 dimensions. Our time series encoder also adopts a 256-dimensional output. Moreover, as demonstrated in Fig. **??**, increasing the embedding dimension beyond a certain point does not yield better performance.

In contrast, LLMs tend to produce significantly higher-dimensional embeddings, with substantial variation across models. For instance, GPT-2 generates 768-dimensional embeddings, while Qwen3-Embedding-0.6B outputs embeddings of size 151,669, as shown in Table. 3. This discrepancy introduces two major challenges: (1) the high dimensionality of LLM embeddings imposes a substantial computational burden; (2) the magnitude imbalance between the two modalities may lead to biased fusion when passed to the causality augmenter.

To address these issues, we employ a Harmonizer—a lightweight projection layer—to map LLM embeddings into the same dimensional space as that of the time series encoder. This design not only reduces computational overhead but also enables a balanced integration of causal signals from both the LLM and the time series encoder, thereby facilitating more robust causal inference.

Specifically, as shown in Eq. (10) in the main text, the embedding output from the pre-trained LLM, denoted as $\widetilde{\mathcal{P}}_\tau \in \mathbb{R}^{N \times M}$, is subjected to a variable-wise linear transformation to project it into a unified representation space:

$$\widetilde{\mathcal{P}}_\tau = \left\{ \widetilde{\mathbf{p}}_{\tau,n} \mathbf{w}_H \mid n = 1, \ldots, N \right\}, \tag{23}$$

where $\widetilde{\mathbf{p}}_{\tau,n} \in \mathbb{R}^M$ represents the embedding of the $n$-th variable obtained by extracting the last token from the LLM output, and $\mathbf{w}_H \in \mathbb{R}^{M \times d}$ denotes the learnable projection matrix used for dimensional alignment.

### A.3.4 CAUSALITY AUGMENTER

Our causality augmenter is built upon the proposed CASA mechanism, where each application of CASA corresponds to constructing a CASA block for enhancing causal inference. We consider two

| Pre-trained LLM | Demision |
|---|---|
| GPT-2 | 768 |
| Qwen3-Embedding-0.6B | 151669 |
| gemma-3-1b | 262144 |

Table 3: Embeddings dimensionality of Pre-trained LLM.

design strategies: (1) constructing a single CASA block and repeating its computation for $1$–$l$ iterations, inferring causal relationships based on the learned parameters $\omega_v$; (2) constructing $l$ parallel CASA blocks, which are jointly optimized using proximal gradient descent, and inferring causality by aggregating their respective $\omega_v$ representations.

In this work, we adopt the first strategy — using only one CASA block and computing it once per iteration — to reduce model complexity. This design choice allows for a more interpretable representation of causal strength via $\omega_v$, while avoiding the potential performance degradation and reduced interpretability that may arise from overly complex multi-block designs. Experimental results further confirm the effectiveness of this streamlined approach.

## A.4 MODEL COMPLEXITY

Compared to conventional methods, our LLM-GC framework introduces a trade-off between semantic enrichment and computational cost. By leveraging the expressive power of large language models (LLMs), LLM-GC achieves notable improvements in Granger causality discovery performance, at the expense of increased model complexity and parameter scale, as summarized in Table 4.

Benefiting from recent advances in computing infrastructure, the training process of LLM-GC can be decomposed into two main stages: (1) generating embeddings from the pre-trained LLM based on designed prompts, and (2) learning the causal structure by integrating LLM-derived embeddings with time-series signals.

The first stage dominates the overall computation, accounting for approximately 87.7% of the total training time. This is due to the fact that embedding generation invokes the full parameter set of the pre-trained LLM, resulting in substantial memory and computation consumption. In contrast, the second stage exhibits comparable efficiency to baseline methods, attributed to our dual-modality encoding design that balances semantic richness and temporal representation.

Overall, while LLM-GC increases the training time from roughly 100 seconds (typical for traditional approaches) to about 1000 seconds, it yields a significant performance gain, improving causality detection accuracy by up to 55%. This trade-off is particularly favorable in high-stakes applications where inference quality outweighs marginal cost increases.

| | Model | Tunable Parameters | LLM Embedding Time (s) | Training time per epoch (s) | Total time (s) | AUROC (↑) |
|---|---|---|---|---|---|---|
| | GC | 10516 | - | 0.014 | 103 | $0.633 \pm 0.026$ |
| | PCMCI | - | - | - | 174 | $0.608 \pm 0.022$ |
| | NGC | 104210 | - | 0.047 | 207 | $0.721 \pm 0.043$ |
| | CR-VAE | 264210 | — | 0.075 | 144 | $0.923 \pm 0.013$ |
| | CUTS | 286640 | - | 0.035 | 142 | $0.721 \pm 0.043$ |
| | KANGCI | 52540 | - | 0.068 | 157 | $0.850 \pm 0.020$ |
| | JRNGC | 3210 | - | 0.015 | 33 | $0.934 \pm 0.018$ |
| **LLM-GC(ours)** | GPT-2 | 124,439,808 | 1298 | 0.582 | 1475 | $0.979 \pm 0.033$ |
| | Qwen-3-Embedding-0.6B | 595,776,512 | 5841 | 0.577 | 1494 | $\mathbf{0.983 \pm 0.042}$ |
| | gemma-3-1b | 999,885,952 | 9682 | 0.588 | 1526 | $0.981 \pm 0.023$ |

Table 4: Comparison of different models in terms of number of tunable parameters, training time per epoch, total training time, and AUROC on Lorenz(20,1000,10).

## A.5 BASELINES

We perform comparative experiments with seven competitive methods: GC (Granger, 1969), PCMCI (Runge et al., 2019), NGC (Tank et al., 2022), CR-VAE (Li et al., 2023), CUTS (Cheng et al., 2023), JRNGC (Zhou et al., 2024), KANGCI (Liu et al., 2025b).

- **GC**: A classical linear autoregressive model that captures temporal dependencies among time series. It serves as the most fundamental Granger causality baseline.

- **PCMCI**: A statistical method based on conditional independence testing, rather than the Granger framework. We follow the original implementation with default test strategies.

- **NGC**: A neural architecture that combines MLPs and RNNs with weight penalties to estimate Granger causality. We adopt its component-wise LSTM variant in our experiments.

- **CR-VAE**: A hybrid model that integrates LSTM with variational autoencoders to jointly perform causal discovery and time series generation.

- **CUTS**: A neural Granger causality method that learns sparse causal graphs by estimating adjacency matrices from imputed time series data.

- **JRNGC**: A unified model that jointly infers both summary and full-time Granger causality across all target variables using a shared encoder-decoder structure.

- **KANGCI**: A nonparametric causal discovery method that adapts Kolmogorov–Arnold Networks (KANs) to learn expressive causal mechanisms from time series.

These baselines cover a diverse range of paradigms, including linear modeling, statistical testing, neural Granger formulations, and generative approaches, providing a comprehensive benchmark for evaluating LLM-GC.

## A.6 METIRCES

### A.6.1 EVALUATION METRICS

To evaluate the performance of Granger causality discovery models, we adopt four standard metrics: AUROC (Area Under the Receiver Operating Characteristic Curve), AUPRC (Area Under the Precision-Recall Curve), F1 Score, and SHD (Structural Hamming Distance).

**AUROC**. The AUROC quantifies a model's ability to distinguish between positive (causal) and negative (non-causal) variable pairs. It is computed by plotting the true positive rate (TPR) against the false positive rate (FPR) at various classification thresholds, and calculating the area under this curve. Mathematically:

$$\text{TPR} = \frac{TP}{TP + FN} \tag{24}$$

$$\text{FPR} = \frac{FP}{FP + TN} \tag{25}$$

AUROC provides a threshold-independent view of ranking quality. However, in sparse settings where negative pairs dominate, it may overestimate performance due to the easy identification of negatives.

**AUPRC**. AUPRC evaluates model performance specifically on the positive (causal) class, making it more suitable for imbalanced settings. It is computed by plotting precision against recall across thresholds and calculating the area under the curve:

$$\text{Precision} = \frac{TP}{TP + FP} \tag{26}$$

$$\text{Recal} = \frac{TP}{TP + FN} \tag{27}$$

In sparse causality scenarios, where the number of true causal edges is small, AUPRC captures a model's ability to retrieve these rare but important relationships. A high AUPRC indicates not only that the model detects true causal links, but that it ranks them above irrelevant ones. As such, AUPRC is a more sensitive and reliable metric for evaluating causal discovery under sparsity.

**SHD**. Structural Hamming Distance (SHD) measures the difference between the predicted and ground-truth causal graphs. It is defined as the number of edge insertions, deletions, or flips re-

quired to convert one graph into the other. Formally:

$$\text{SHD}(G, \hat{G}) = \sum_{i,j} [G_{ij} \neq \hat{G}_{ij}] \tag{28}$$

SHD directly reflects structural accuracy, offering an interpretable way to assess how closely the inferred graph matches the true causal structure. This is particularly important for downstream applications that depend on correct edge-level reasoning. In sparse settings, lower SHD indicates that the model avoids introducing spurious edges while successfully recovering the limited true ones, thus balancing precision and completeness at the graph level.

**F1 Score**. The F1 score provides a balanced evaluation of a model's performance by combining precision and recall into a single metric. It is particularly useful when both false positives and false negatives carry significant implications. The F1 Score is defined as the harmonic mean:

$$\text{F1} = 2 \cdot \frac{\text{Precision} \cdot \text{Recall}}{\text{Precision} + \text{Recall}} = \frac{2 \cdot TP}{2 \cdot TP + FP + FN} \tag{29}$$

In sparse Granger causality settings, the F1 Score captures the model's ability to recover true causal edges (recall) while minimizing false positives (precision). Unlike AUROC, it is threshold-dependent and reflects performance in binary causal graph predictions. A high F1 Score indicates a balanced trade-off between sensitivity and specificity, making it a practical and interpretable metric for evaluating structural accuracy.

### A.6.2 MAIN METRIC AND MOTIVATION: AUORC

In causal graph inference, AUROC (Area Under the Receiver Operating Characteristic Curve) is widely adopted for its robustness and threshold-independence. It jointly considers the true positive rate and false positive rate across all thresholds, offering an objective measure of the model's ability to distinguish between causal and non-causal relationships.

AUROC is particularly valuable for two reasons: (1) It balances recall and precision, aligning well with the goal of structural accuracy in causal discovery. (2) It remains robust under severe class imbalance—common in high-dimensional or sparse graphs (e.g., gene networks or brain connectivity)—where traditional metrics like accuracy may be misleading.

In summary, AUROC provides a comprehensive and fair evaluation of how well the inferred causal graph aligns with the ground truth, making it a reliable benchmark for comparing causal discovery methods, especially in complex real-world systems.

### A.7 DATASET DETAILS

We evaluate the proposed LLM-GC framework on five publicly available benchmark datasets, which are widely adopted in recent studies and serve as standard baselines for evaluating Granger causality discovery methods.

Compared to synthetic datasets such as VAR and Lorenz-96, which are generated from predefined equations with explicit dynamical mechanisms, real-world datasets typically lack known underlying dynamics or functional relationships. As a result, causal discovery on real data is inherently more challenging due to noise, complexity, and potential confounding factors.

### A.7.1 VAR DATASET.

The vector autoregressive (VAR) model is a classical linear multivariate time series model widely used for causal structure discovery and forecasting. It is defined as:

$$\mathbf{x}_t = \sum_{\alpha=1}^{\tau} A_\alpha \mathbf{x}(t - \alpha) + \boldsymbol{\varepsilon}(t), \tag{30}$$

where $\mathbf{x}_t \in \mathbb{R}^N$ denotes a $N$-dimensional multivariate time series at time $t$, $\tau$ is the maximum time lag, and $\boldsymbol{\varepsilon}(t)$ is a zero-mean noise term, often assumed to follow a Gaussian distribution. Each matrix $A_\alpha \in \mathbb{R}^{N \times N}$ captures the linear dependencies between variables at lag $\alpha$.

In our experiments on synthetic VAR datasets, we denote a configuration as **VAR**($N$**,** $T$**,** $\tau$), where $D$ is the number of time series (variables), $T$ is the total number of time points, and $\tau$ is the true time lag. For example, **VAR(20, 1000, 5)** represents a setting with 20 variables, 1000 time steps, and a true time lag of 5.

### A.7.2    LORENZ-96 DATASET.

The Lorenz-96 model is a canonical chaotic dynamical system widely used to evaluate the performance of time-series forecasting and causality inference algorithms due to its controllable complexity and rich nonlinear dynamics (Karimi & Paul, 2010). Originally proposed to mimic atmospheric energy transport, the model captures essential characteristics of spatiotemporal chaos, including sensitivity to initial conditions and multiscale interactions among variables.

The model is defined as:

$$\frac{dx_{t,i}}{dt} = (x_{t,i+1} - x_{t,i-2}) \, x_{t,i-1} - x_{t,i} + F, \tag{31}$$

where $x_{t,i}$ denotes the state of the $i$-th variable at time $t$, and $F$ is a constant external forcing term that governs the degree of chaos in the system. For instance, when $F = 20$, the system exhibits strong chaotic behavior.

The index $i$ is defined modulo $N$, i.e., $x_{-1} = x_{N-1}$, $x_0 = x_N$, and $i = 1, 2, \ldots, N$, which enforces periodic boundary conditions. This structure allows the Lorenz-96 model to simulate a closed ring of interacting variables, making it suitable for testing Granger causality discovery in high-dimensional and nonlinear settings. In our experiments on Lorenz-96 datasets, we denote **Lorenz(20, 1000, 10)** to represent a scenario where there are 20 dimensions, 1000 time points in total, and the chaotic behavior $F$ is set to 10.

### A.7.3    FMRI DATASET.

(Smith et al., 2011) generated rich, realistic simulated fMRI data for a wide range of underlying networks, experimental scenarios, and problematic confounders in the data to compare different approaches to connectivity estimation. Each data includes multiple time series corresponding to different brain regions of interest (ROIs) using the dynamic causal model (DCM) with the nonlinear balloon model for vascular dynamics. It is publicly available and usually used to estimate the brain network. The dataset consists of 50 subjects, with each subject having 15 nodes and 200 observations.

### A.7.4    DREAM-3 DATASET.

The DREAM-3 dataset is a publicly available realistic gene expression data set from the DREAM-3 challenge (Prill et al., 2010), mentioned in many causal discovery works as quantitative benchmarks. This challenge includes five simulated datasets, comprising two E. coli datasets and three yeast datasets, each featuring a distinct underlying Granger causality plot. Each dataset contains 10 numbers of different time series, each with 4 replicates, sampled at 21 time points, resulting in a total of 966 time points. As can be seen, the data is very limited in length and is a difficult nonlinear dataset.

### A.7.5    DREAM-4 DATASET.

The DREAM-4 network inference challenge, introduced by (Marbach et al., 2010) and publicly available, aims to facilitate the reconstruction of gene regulatory networks from gene expression time-series data. The challenge includes five independent datasets, each consisting of ten time-series recordings that track the expression levels of 10 genes across 21 time steps, each with 5 replicates.

### A.7.6    SUMMARY OF DATASETS.

Among the five datasets used in our experiments, DREAM-4 poses the greatest challenge due to its high dimensionality, limited number of observations, and unknown underlying dynamical mechanisms. As such, it serves as a rigorous benchmark to assess the overall robustness and generalization capability of causality discovery methods.

```
{
  "instruction":"
    You are an expert in time series representation learning algorithms.
    Given the historical observations of time series along with background (contextual) information, the task is to
infer the Granger causal relationships among the variables.
  ",
  "input":"
    [Dataset Context]: <This dataset represents gene expression levels, where the network topologies were from the
transcriptional regulatory networks of E. coli and S. cerevisiae. It includes 100 genes over 100 time points. >
    [Historical Data]: From <τ−L > to <τ−1>, the values were <x_(τ−L),…,x_(τ−1)> every f
    [Statistical Features]: The overall trend is <Δ_(τ,i)>, the median is <median_val>…
    [Task Instruction]: <Please comprehensively represents the time series>
  ",
  "output":"
    Time series <1> Granger-causes <2>,
    Time series <3> Granger-causes <2>."
}
```

Figure 8: An example instruction triplet (*instruction*, *input*, and *output*) used for fine-tuning the LLM on Granger causality discovery.

## A.8 IMPLEMNETATION EXPERIMENT DETAILS

### A.8.1 FINE-TUNING LLM FOR GRANGER CAUSAL DISCOVERY

To evaluate the effectiveness of fine-tuning large language models (LLMs) for Granger causal discovery, we construct a dedicated instruction-tuning corpus consisting of *instruction*, *input*, and *output* triplets, as shown in Fig. 8.

The **instruction** component specifies the task goal—namely, to infer Granger causal relationships—thus guiding the LLM to activate domain-relevant capabilities. The **input** contains historical time series data along with contextual background information, encouraging the model to extract causal cues and key temporal patterns. To ensure fair comparison, the content and structure of the input are aligned with those used in the LLM-GC framework, with the only modification being that historical data is presented at the multivariate level (i.e., all variables simultaneously), rather than variable-wise, to enable system-wide causality detection. The **output** serves as the ground-truth causal graph, providing supervision for training loss and aligning model predictions with true underlying dependencies.

From an implementation perspective, we utilize the `llama-factory` toolkit (Zheng et al., 2024) for seamless fine-tuning. Specifically, we adopt the Low-Rank Adaptation (LoRA) technique (Hu et al., 2022), which injects trainable low-rank matrices into the Transformer layers of the LLM. This allows efficient adaptation to downstream tasks—such as causal discovery—without incurring the full cost of end-to-end parameter updates.

LoRA operates by freezing the pre-trained weights $\mathbf{W} \in \mathbb{R}^{d \times k}$ and learning a low-rank update matrix through decomposition:

$$\Delta\mathbf{W} = \mathbf{BA}, \tag{32}$$

$$\mathbf{B} \in \mathbb{R}^{r \times k},\ \mathbf{A} \in \mathbb{R}^{d \times r},\ r \ll \min(d, k) \tag{33}$$

The modified forward computation becomes:

$$\mathbf{h} = \mathbf{W}x + \alpha \cdot BAx \tag{34}$$

$$\alpha = \frac{1}{r} \tag{35}$$

During training, LoRA performs the following: (1) freezes all original model parameters, (2) injects rank-$r$ adapter pairs $(\mathbf{A}, \mathbf{B})$ into the query and value projection matrices of each Transformer layer, and (3) updates only the adapter weights. This results in a memory footprint reduction of approximately 75%, significantly lowering training overhead while preserving model performance. To ensure fair comparison and consistency, we use the same fine-tuning hyperparameter configuration for all datasets, as shown in Table 5.

| Hyperparameters | Starting value |
|---|---|
| Learning rate | $2 \times 10^{-4}$ |
| Batch size | 16 |
| Dropout | 0.1 |
| Epoch | 10 |
| Rank | 8 |
| $\alpha$ of LoRA | 16 |
| Dropout of LoRA | 0.1 |
| Rate of LoRA+LR | 4 |

Table 5: Hyperparameter configurations for fine-tuning the LLM for Granger causal discovery using the `llama-factory` platform.

### A.8.2 MORE DETAILS FOR EXPERIMENT FOR DREAM-3 AND DREAM-4 DATASETS

Due to the inherent characteristics of the dataset, self-causality (i.e., a variable causing itself) is not considered during data collection. Therefore, in our experiments, when modeling the causal parents of a target time series $i$, we exclude $i$ itself from the input. Specifically, the model is trained using the remaining 9 time series, thereby constraining the discovery of Granger causality to inter-variable relationships only.

### A.9 EXPERIMENTAL HYPERPARAMETERS

For each experiment conducted with LLM-GC, we begin by initializing a consistent set of hyperparameters. During training, certain parameters—such as the learning rate—are adaptively adjusted based on the model's performance in fitting the underlying dynamics of the time series data. Table 6 presents the optimal initialization settings that yielded the best empirical performance across datasets.

To further promote model sparsity and enhance generalization, we apply an $\ell_2$-norm regularization term with a weight of 0.1 to the entire network, thereby penalizing excessive parameter magnitudes during optimization.

| Hyperparameters | Starting value |
|---|---|
| Learning rate | $5 \times 10^{-4}$ |
| Batch size | 64 |
| Dropout | 0.1 |
| Epoch | 1000 |
| Dimension of attntion & CASA | 256 |
| Dimensions of Feed-forward Network | 32 |
| Encoder layer | 1 |
| Proximal step size $\rho$ | 9.5 |
| Trade-off $\lambda$ | 0.1 |
| Early stop epoch patience | 50 |
| Network regularizer | 0.1 |

Table 6: Initialization of hyperparameters for LLM-GC at the beginning of experiments.

### A.10 ADDITIONAL EXPERIMENTAL RESULTS

### A.10.1 OVERALL PERFORMANCE

We visualize the performance of LLM-GC across all five benchmark datasets, as illustrated in Fig. 9. For each benchmark, a representative subsetting of the dataset is selected to demonstrate the evaluation results.

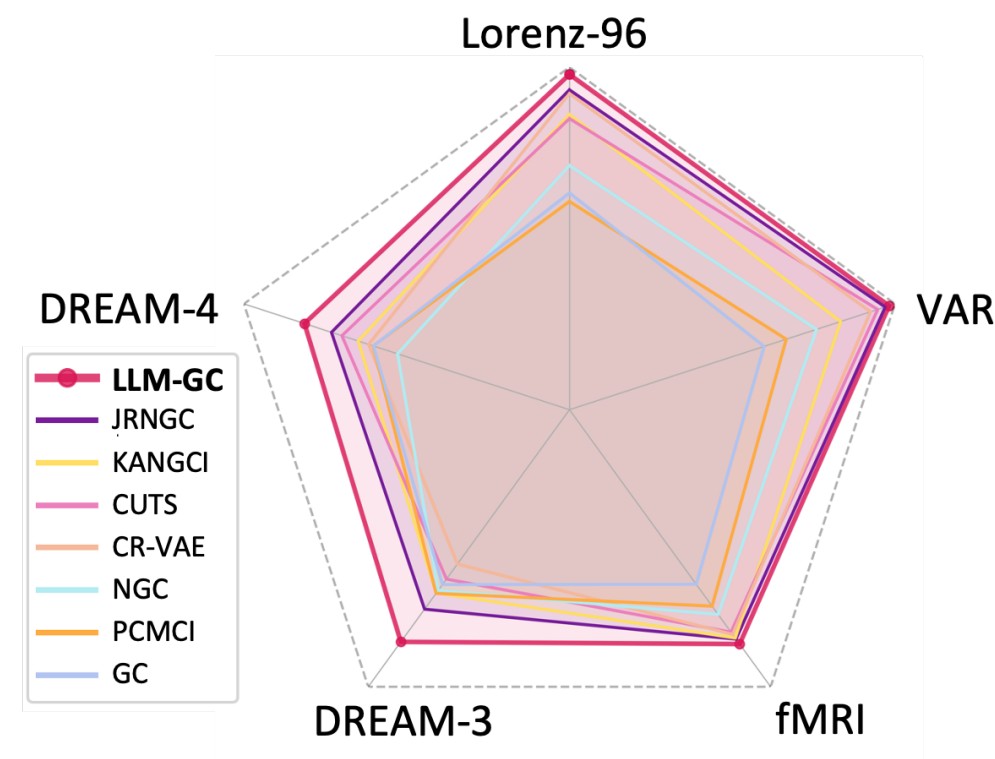

Figure 9: Overall performance of LLM-GC on five benchmarks compared to baselines.

As shown in Fig. 9, LLM-GC consistently achieves superior performance across all five benchmark datasets. It significantly outperforms traditional Granger causality discovery methods (e.g., GC, PCMCI, NGC) and demonstrates competitive advantages over recent learning-based approaches such as CUTS and JRNGC. Notably, the improvement is particularly evident on complex datasets such as DREAM-3 and DREAM-4, highlighting the benefit of integrating LLM-driven semantic signals with time series dynamics.

### A.10.2 ABLATION STUDY ON PROMPTS

We compare the impact of different prompt designs on LLM-GC performance, with a particular focus on *statistical features*. As shown in Fig. 10, we evaluate four variants of statistical cues embedded in the prompt: (1) **w/o**, i.e., no statistical information; (2) **Average**, which includes the mean value of the variable's history; (3) **Trend**, which reflects the overall temporal change direction; (4) **Review**, which encodes the total number of historical hours observed.

Among these variants, incorporating **trend** features yields the best AUROC performance, suggesting that directional dynamics are particularly helpful for the LLM to infer causality. Including **average** or **review** statistics also leads to slight improvements over the baseline without statistics, demonstrating the general benefit of injecting structured numerical context. Notably, all variants that incorporate statistical features outperform the **w/o** setting, confirming the importance of lightweight numerical summarization in enhancing semantic prompts for causal discovery.

### A.10.3 ABLATION STUDY ON PRE-TRAINED LLM FOR EMBEDDING GENERATION

We further investigate the impact of using different pre-trained LLMs for generating prompt embeddings in the LLM-GC framework. While our method is model-agnostic and can accommodate a wide range of LLMs, different models may vary in representation quality, embedding dimensionality, and alignment capability with time-series dynamics.

**[Statistical Features]:** The overall trend is $<\Delta_{\tau,i}>$

P1: abstract temporal trends

**[Statistical Features]:** The total number of historical hours is $<96>$

P2: review historical time

**[Statistical Features]:** The average value is $<\Delta_{\tau,i}>$

P3: summarize average value

**[Statistical Features]:**

P4: without statistical features

Figure 10: Ablation study on different types of statistical features included in LLM prompts. Each variant represents a different prompt design, and the resulting AUROC is shown. All statistical prompts improve performance compared to the baseline without statistics (w/o).

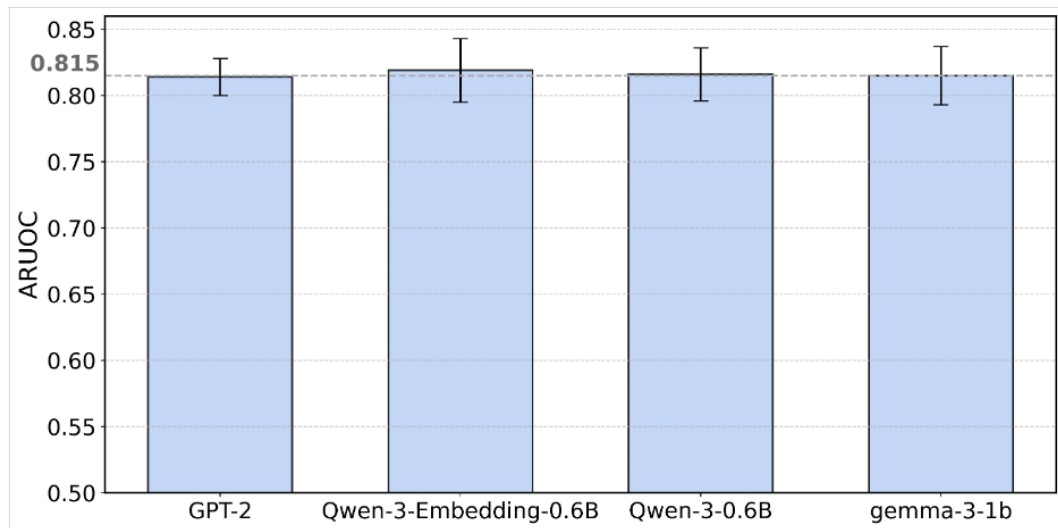

Figure 11: Ablation on different pre-trained LLMs used for prompt embedding generation in LLM-GC. All models achieve consistent ARUOC performance, validating the generality of the framework.

As shown in Fig. 11, we compare four representative pre-trained LLMs: (1) **GPT-2**, (2) **Qwen-3-Embedding-0.6B**, (3) **Qwen-3-0.6B**, and (4) **Gemma-3-1B**.

All models achieve consistent ARUOC performance in the range of 0.80–0.82, indicating the general robustness of our framework across different LLM backbones. Among them, **Qwen-3-Embedding-0.6B** performs best, slightly outperforming others, which may be attributed to its optimized structure for embedding extraction.

Interestingly, even smaller-scale models such as GPT-2 and Qwen-3-0.6B yield competitive results, suggesting that massive model size is not strictly necessary for capturing causal-relevant semantics. This supports the efficiency and scalability of LLM-GC, making it suitable for low-resource or latency-sensitive scenarios.

These findings show that the strength of LLM-GC lies in the prompt-driven design and dual-modality fusion, rather than relying on the scale or specific architecture of the underlying LLM.

### A.10.4 VISUALIZATION OF FOUR EMBEDDINGS

To better understand the learned representations in LLM-GC, we visualize four types of embeddings using t-SNE (top) and PaCMAP (bottom), as shown in Fig. 12. The embeddings include: (1) **TS emb** — representations from the time series encoder, (2) **Prompt emb** — representations generated by the LLM from prompt inputs, (3) **Adding TS and P** — the additive fusion of both modalities, and (4) **Dual-retrieval** — our proposed retrieval-enhanced fusion representation.

In both projections, we observe that the **TS embeddings** (yellow) form a dense cluster with limited inter-variable separation, indicating a strong temporal coherence but weak semantic distinction. In contrast, the **Prompt embeddings** (orange) are more dispersed and exhibit richer inter-variable structure, capturing higher-level semantics beyond pure temporal signals.

The fused representation from **Adding TS and P** (purple) shows moderate improvement in alignment but still suffers from inter-cluster blending. Notably, our proposed **Dual-retrieval** embedding (red) forms clearly separated and well-structured clusters in both visualizations. This suggests that the dual-modality retrieval mechanism enhances semantic grouping while maintaining temporal relevance, leading to improved causal signal disentanglement.

These observations support our claim that aligning temporal and prompt semantics through dual retrieval yields more structured, interpretable, and task-relevant representations for Granger causal discovery.

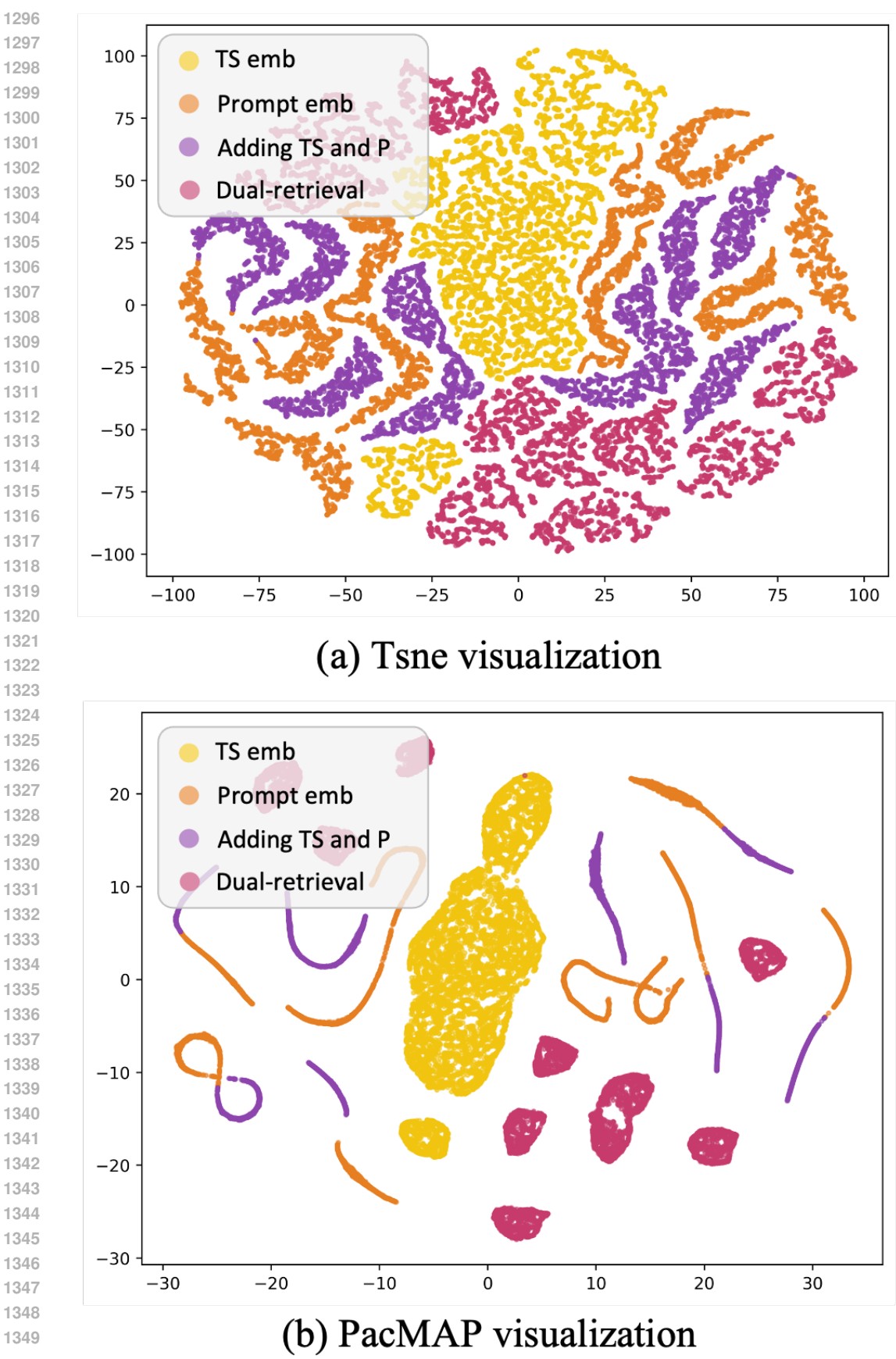

(a) Tsne visualization

(b) PacMAP visualization

Figure 12: Visualization of four types of embeddings using t-SNE (top) and PaCMAP (bottom). Dual-retrieval embeddings exhibit clearer inter-variable separation and semantic structure, validating their effectiveness in multi-modal alignment.

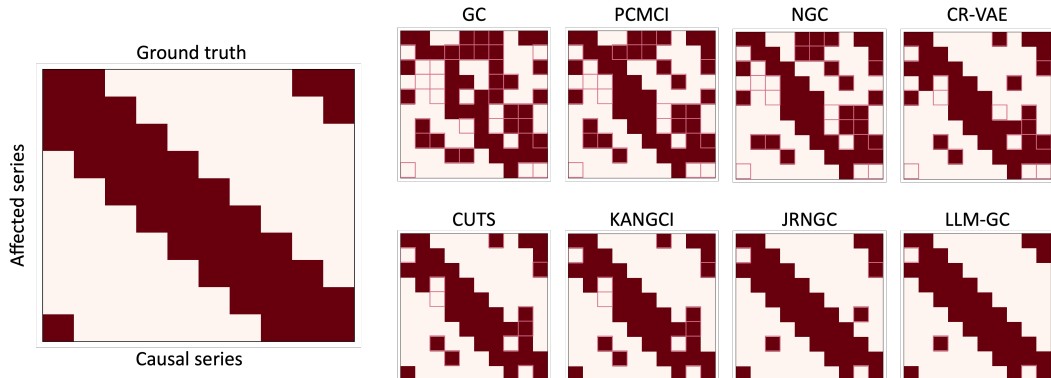

Figure 13: Causal graph predictions on the Lorenz-96(20,1000,20) dataset. Left: ground-truth graph; Right: inferred graphs by different methods. Red boxes denote predicted edges; empty grids indicate ground-truth links. LLM-GC best approximates the sparse local interaction pattern.

### A.10.5 VISUALISATION OF CAUSAL GRAPH

Fig. 13 presents a visual comparison of the learned Granger causal graphs from different methods on the Lorenz-96(20,1000,20). The ground-truth causal structure is shown on the left, characterized by a sparse banded pattern that reflects local interactions among neighboring variables.

The remaining subplots display the inferred causal graphs from baseline methods, where red squares denote predicted causal edges and white grids indicate ground-truth positions. We observe the following:

**LLM-GC** (bottom-right) produces a causal graph that most closely resembles the ground truth, accurately capturing both the overall sparse structure and the local connectivity patterns. It shows minimal false positives and aligns well with the known dynamics of Lorenz-96.

**Traditional neural methods** such as NGC, CR-VAE, and JRNGC tend to generate dense or noisy graphs with significant false positives, often failing to reflect the localized interaction structure.

**Statistical methods** such as PCMCI and GC either underfit or overfit the structure, leading to scattered or missing edges.

This visualization confirms that LLM-GC benefits from both temporal encoding and semantic alignment, enabling it to recover interpretable and faithful causal structures even in chaotic dynamical systems.

