# OpenReview forum: "LLM-GC: Temporal-Semantic Disentanglement with Retrieval Augmentation to Activate LLM's Ability for Multimodal Granger Causal Discovery"
_ICLR.cc/2026/Conference — ICLR 2026 Conference Withdrawn Submission_

### Official Review · Reviewer_N59e · 2025-10-31

**Soundness:** 2
**Presentation:** 2
**Contribution:** 2
**Rating:** 2
**Confidence:** 4

**Summary:**

LLM-GC is a multimodal framework for Granger causal discovery that augments time-series data with frozen LLM semantic priors. For each variable, a textual prompt is encoded by the LLM and projected via a Harmonizer, then Cross-Modal Dual Retrieval (DR) performs bidirectional variable-wise attention to selectively inject helpful textual cues into the time-series representation under a forecasting loss. From this fused representation, a Causality Augmenter with CASA (variable-axis self-attention with sparsity) yields an interpretable N×N Granger adjacency capturing directed, conditional influences. Across synthetic (VAR, Lorenz-96) and real benchmarks (fMRI, DREAM), LLM-GC improves AUROC/AUPRC/F1 and reduces SHD versus classical/neural baselines; benefits depend on prompt relevance and challenges remain for hidden confounding and precise lag identification.

**Strengths:**

- Introduces a multimodal GC framework (LLM-GC) that fuses variable-wise time-series encoding with LLM-derived semantic priors via cross-modal dual retrieval—explicitly aligning temporal and contextual signals for causal discovery.
- Proposes Causality-Aware Self-Attention (CASA) to emphasize variable-to-variable, effect-driven patterns, yielding more interpretable graphs and state-of-the-art results across synthetic and real-world benchmarks.

**Weaknesses:**

- The paper states that Dual Retrieval (DR) aligns/fuses time-series and text, and then CASA consumes the fused representations (or features derived from them) to produce the Granger adjacency matrix. However, the figure depicts the opposite order (CASA followed by DR), which is confusing. Clarification on the actual computation flow is needed.
- The Harmonizer module is referenced but not clearly described in the main text. It is only vaguely mentioned as appearing in the appendix, yet no explicit figure/section pointer is provided. In addition, the sentence “as demonstrated in Fig. ?? …” suggests missing or incorrectly referenced figures, making the role of Harmonizer and its empirical evidence unclear.

**Questions:**

- It is ambiguous whether CASA receives inputs from each modality separately, or a joint representation of both modalities. The authors should clarify the exact input interface of CASA and whether parameters are shared across modalities or sequentially applied.
- An ablation or baseline without prompts (pure TS-based GC) is necessary to isolate the contribution of textual priors. Showing performance of CASA alone (TS-only) would help validate that text actually provides added value rather than simply restating information contained in the time series.

---

### Official Review · Reviewer_VSKC · 2025-11-01

**Soundness:** 2
**Presentation:** 2
**Contribution:** 2
**Rating:** 2
**Confidence:** 3

**Summary:**

The paper proposes LLM-GC, a multimodal framework for Granger causal discovery in time series, integrating temporal encodings with LLM-derived semantic prompts via variable-wise dual-modality encoding, cross-modal retrieval alignment, and causality-aware self-attention (CASA) to infer graphs from datasets like VAR, Lorenz-96, fMRI, and DREAM-3/4. While claiming superior AUROC/SHD over unimodal baselines, the approach relies on opaque prompt construction and domain-specific knowledge injection, making claims unverifiable.

**Strengths:**

- One of the earliest attempts combining LLMs in Granger causal discovery and time-series forecasting.

- Demonstrates the potential of LLM as a superior feature extractor of real-valued time-seris.

- Provides a framework for cross-modal TS and language alignment, which could be useful in other applications.

**Weaknesses:**

Prompt "wrapping" process is quite unclear, vaguely described as dynamically instantiating templates with domain contexts (e.g., "gene expression levels" for DREAM) without specifying functions for serialization or stats computation (e.g., trend formula), leading to unreproducible metadata insertion and potential tautology in causal inference.​

Comparisons could be unfair, pitting LLM-GC's knowledge-enriched prompts (human-provided domain priors) against unimodal baselines (GC, PCMCI, etc.) without equivalent semantic augmentation, biasing results, effectively "pre-answering" causality via hints like Lorenz-96 forcing F=10.​

**Questions:**

The paper doesn't clearly describe how the base template was prepared in a dataset-specific manner. Please clarify.

---

### Official Review · Reviewer_Z3TQ · 2025-11-01

**Soundness:** 3
**Presentation:** 3
**Contribution:** 2
**Rating:** 4
**Confidence:** 3

**Summary:**

The paper proposes a novel framework called LLM-GC, which integrates LLMs with time-series analysis to improve Granger causal discovery (GCD). Traditional GCD methods rely solely on raw temporal data, which limits contextual understanding and leads to overfitting. LLM-GC overcomes these issues by introducing a dual-modality system: one branch models temporal dynamics using a time-series encoder, and the other encodes domain and contextual semantics via prompts processed by a pretrained LLM. These representations are aligned through a cross-modal dual retrieval mechanism, and causal relationships are inferred using a Causality-Aware Self-Attention (CASA) module that enhances interpretability by focusing on variable-level dependencies.

**Strengths:**

- This is the first framework to systematically connect large language models with Granger causal discovery, moving beyond unimodal temporal modeling to a multimodal paradigm.
- The authors redefine GCD by introducing textual semantic priors as an auxiliary modality, expanding the conventional time-series setup to a multimodal causal discovery problem.

**Weaknesses:**

- While the proposed CASA mechanism is conceptually aligned with variable-level dependencies, the paper does not provide qualitative evidence or case studies demonstrating how CASA attention maps correspond to interpretable causal patterns. The current evidence is primarily quantitative (AUROC, SHD).
- The paper’s Definition 1 extends Granger causality to include textual priors but does not provide a formal proof that such semantic terms maintain the original Granger assumptions (e.g., temporal precedence, conditional independence).
- The framework heavily depends on manually designed prompt templates (Fig. 3) and a frozen LLM (GPT-2). These prompts encode dataset context and statistical features, but their design seems heuristic and domain-specific. The paper does not explore robustness to prompt variations or automated prompt generation.
- The baseline set includes unimodal Granger causal discovery models but omits comparisons with general multimodal or retrieval-augmented architectures (e.g., CLIP-like models or cross-modal transformers used in multimodal alignment).
- The paper reports uniformly high results but omits analysis of where LLM-GC fails—e.g., cases with noisy prompts, missing modalities, or low data availability.

**Questions:**

- In Definition 1 (page 3), the paper introduces “Multimodal Granger Causal Discovery” by incorporating textual prompts $P_\tau$ into the causal condition. However, it remains unclear how this extension preserves key properties of classical Granger causality—specifically temporal precedence and conditional independence. Could the authors provide a short proof sketch or intuitive justification that multimodal GC reduces to standard GC when $P_\tau$​ is absent or independent of $X_\tau$? This would help validate that the framework maintains causal interpretability rather than introducing semantic correlations.

---

### Official Review · Reviewer_EstF · 2025-11-01

**Soundness:** 2
**Presentation:** 2
**Contribution:** 2
**Rating:** 4
**Confidence:** 4

**Summary:**

The authors proposed a retrieval augmented approach to use LLMs for granger causal discovery. Granger causality is a statistical concept used to determine if one time series can predict another. It is based on the idea that if variable X causes variable Y, then the past values of X should provide significant information about the future values of Y. However, from Definition 1, it defines Granger non-causal using Eq (2), which is very hard to understand how this is correlated to Granger causality.

One of the contributions of the paper is the "Causality-Aware Self-Attention". However, that module is simply a variable level attention, and it would be very hard to view the relation between the attention module and causality. Very likely, that module is still based on the correlation between variables. I guess, the final model is with moth temporal attention and variable level attention, along with the regularizers in (17).

Experiments on both synthetic and real dataset demonstrates the proposed approach somehow works.

**Strengths:**

1. The idea of using LLMs for granger causal discovery is somehow novel.
2. The experimental results seems to be good.

**Weaknesses:**

1. The definition of granger causality in Definition is very unclear and very hard to understand. Instead of define Granger non-causality, it would be better to define Granger causality.

2. We can hardly align (17) with granger causality. If we would like to find the granger parents of a variable i, then we will need to look into the past variables that provides most information for prediction of i, that is by removing some variable j we can not accurately predict variable i. However, the author failed to demonstrate that the current model can do this.

3. There are some minor errors e.g. line 785 Fig?? In Table 3. are the embedding size for  Qwen3-Embedding-0.6B and gemma-3-1b correct?

**Questions:**

See above weakness.

---

### Note · Authors · 2025-11-16

I have read and agree with the venue's withdrawal policy on behalf of myself and my co-authors.